# Ionic thermoelectric gating organic transistors

Dan Zhao[1], Simone Fabiano[1], Magnus Berggren[1] & Xavier Crispin[1]

Temperature is one of the most important environmental stimuli to record and amplify. While traditional thermoelectric materials are attractive for temperature/heat flow sensing applications, their sensitivity is limited by their low Seebeck coefficient ($\sim 100\,\mu V\,K^{-1}$). Here we take advantage of the large ionic thermoelectric Seebeck coefficient found in polymer electrolytes ($\sim 10,000\,\mu V\,K^{-1}$) to introduce the concept of ionic thermoelectric gating a low-voltage organic transistor. The temperature sensing amplification of such ionic thermoelectric-gated devices is thousands of times superior to that of a single thermoelectric leg in traditional thermopiles. This suggests that ionic thermoelectric sensors offer a way to go beyond the limitations of traditional thermopiles and pyroelectric detectors. These findings pave the way for new infrared-gated electronic circuits with potential applications in photonics, thermography and electronic-skins.

[1] Laboratory of Organic Electronics, Department of Science and Technology, Linköping University, Norrköping SE-60174, Sweden. Correspondence and requests for materials should be addressed to S.F. (email: simone.fabiano@liu.se) or to X.C. (email: xavier.crispin@liu.se).

Among the various environmental stimuli, heat is an important parameter to record and amplify. The temperature sensor/detector can be in direct physical contact or in proximate to the object of interest. The latter case is specific to the field of photonics since heat propagates to the sensor through infrared radiation that changes the temperature of the sensor. Thermal infrared sensors/detectors and infrared cameras find many applications in medical, industrial, military and consumer products[1]. Precision medical thermal imaging of human skin[2] can provide clinically relevant information about pathological conditions that affect how the body regulates temperature, or track a healing process around a scar. Electronic skin (or e-skin[3]) comprises a network of flexible sensors that can conformally wrap irregular surfaces and spatially map and quantify various stimuli[4], which is either used on the human body for medical applications or on machine's body for robotics. Temperature sensors can be bolometer/thermistors[5,6], diodes[5], thermoelectric/thermopiles[7] or pyroelectric sensors[8].

Thermopiles and pyroelectric detectors do not require an electrical input signals (unlike bolometers), but rather develop a large voltage under infrared light illumination. A pyroelectric detector functions similar to a capacitor with the pyroelectric material as the dielectric[9]. The advantage of using a pyroelectric material is the large voltage produced per amount of incident infrared light ($10^4$ V/W)[10]; which makes such devices very sensitive even at room temperature. Pyroelectric detectors have, however, two major disadvantages that limit their commercial applications to detection (for example, burglar alarms) rather than thermometry and thermography. First, under constant exposure, the voltage signal declines with time making it difficult to obtain accurate static temperature measurement without additional expensive mechanical and electronic devices[11]. Second, it is sensitive to vibration since pyroelectric materials also are piezoelectric.

Thermoelectric materials are attractive for temperature/heat flow sensing applications. In thermoelectric effect, charge carriers thermo-diffuse in a material under a temperature difference. This produces a voltage $V = S\Delta T$ proportional to the temperature difference multiplied by the Seebeck coefficient $S$ of the material. The Seebeck coefficient of electronic materials used in thermopiles is typically of the order of a few $100\,\mu V\,K^{-1}$ (ref. 7). Nowadays with microtechnology, thermocouples can be replicated and coupled in series to enhance the sensitivity and reach 100 V/W[12]. The advantage of thermoelectric detection is that the voltage signal is constant versus time under constant illumination; which makes it appropriate for temperature measurement. The drawback is the low Seebeck coefficient of the electronic materials that limits their sensitivity compared with pyroelectric detectors. Recently, inspired by nature[13,14], there has been a renew interest in the thermoelectric properties of electrolytes. When an electrolyte is in fact subjected to a temperature gradient, it undergoes molecular reorganization with formation of an ionic concentration gradient via thermo-diffusion (that is, Soret effect). This results in a large ionic Seebeck coefficient as high as several mV $K^{-1}$, as recently reported for some organic electrolytes[15]. Polymeric electrolyte based on NaOH-treated polyethyleneoxide have also been shown to reach ionic Seebeck coefficients as large as $11\,mV\,K^{-1}$, allowing for charging ionic thermoelectric supercapacitors (ITESCs) by the ionic thermoelectric effect[16].

Potentiometric signals from sensors can be amplified by organic transistors. These are used in fact as smart sensors with the common strategy to shift the potential of the gate through the interaction with an analyte or by external stimuli[17–19]. One of the primary methods to improve sensitivity and selectivity of the transistor-based sensors for probing small changes in the gate bias is to develop devices that operate at low voltages and with high transconductance[20]. In this respect, electrolyte-gated transistors offer unique properties such as low-voltage operation ($<1$ V) even for thick electrolyte layers[21], which is useful in low-power electronics[22], biosensors[23] and smart textile[24,25].

Here we develop an external heat-gated organic transistor, which consists of an electrolyte-gated transistor and an ITESC. When the ionic thermoelectric leg is subjected to a temperature gradient $\Delta T$, it generates a large Soret-induced open voltage that gates the transistor. Hence, this ionic thermoelectric-gated transistor converts a modulation in $\Delta T$ to a modulation in the drain current $\Delta I$. Since the operating voltage of our electrolyte-gated transistors is of the same order of magnitude as the variation of the thermal voltage generated by the ITESC, we are able to demonstrate that it is possible to tune the transistor output current over more than two orders of magnitude with a temperature gradient. This is the first demonstration of thermoelectric gating of a transistor, and it is likely to have important applications in smart sensors, human-organic electronic technology interfacing (that is, e-skin), and active flexible electronics. By coupling the electrical conductivity with thermoelectricity, a new field of thermoelectronics is proposed.

## Results

**Electrolyte-gated transistors**. First, the electrical properties of an electrolyte-gated transistor are characterized. Bottom contact, top gate transistors are fabricated using regioregular poly(3-hexylthiophene-2,5-diyl) (P3HT) as the active semiconducting layer and poly(vinylphosphonic acid-*co*-acrylic acid) (P(VPA-AA)) as the polyanionic electrolyte insulator (Fig. 1a). The P(VPA-AA) phosphonic acid groups are strongly acidic (the first acidic constant, p$K_{a1}$, is about $2.5 \pm 0.5$ in water), and therefore provide plenty of potentially mobile cations (ca. 8 mmol $g^{-1}$, considering one dissociated proton per phosphonic acid group). The anionic phosphonate groups are instead virtually immobile, preventing penetration of anions into the conjugated polymer when a negative gate is applied, and thus precluding electrochemical doping of the bulk semiconducting layer[26]. When a negative bias voltage is applied to the gate, protons in the electrolyte insulator layer drift towards the electrolyte-metal gate interface, while negative immobile ions accumulate at the electrolyte-semiconductor interface, establishing two EDLs (Helmholtz layer). The charged sheets within these Helmholtz layers are separated by only a few Angstroms, leading to a very large gate capacitance. The typical output characteristics of a P3HT-based electrolyte-gated transistor at four different gate voltages ($V_g$) are reported in Fig. 1b. The device displays a large gate modulation of the drain current ($I_{ds}$), in both the linear and saturation regimes, for driving voltages of only a few hundreds of millivolts. The transfer characteristics reported in Fig. 1c, clearly show a large modulation of $I_{ds}$ with on-to-off current ratios typically greater than $10^4$ for $V_g$ range of only 1.5 V, while the threshold voltage ($V_{th}$) is $-0.18$ V. Importantly, drain currents as large as 4 $\mu$A (channel length $L = 10\,\mu m$, channel width $W = 1$ mm) are measured at very low gate and drain biases (that is, $V_g = V_{ds} = -1.2$ V). This low-voltage, high-current operation reflects the very large capacitance of the electrolyte gate dielectric, which induces $>10^{13}$ carrier $cm^{-2}$ in the source-drain channel at gate biases of $<1$ V (see Supplementary Fig.).

**Ionic thermoelectric voltage generators**. As a next step, the thermoelectric properties of the PEO-NaOH electrolyte is investigated using the procedure described before[16]. When NaOH is added to a low molecular weight PEO (Mw = 400 g $mol^{-1}$), the alcohol end-groups (–COH) transform into negatively charged alkoxide end-groups (–CO$^-$ Na$^+$). The resulting electrolyte is

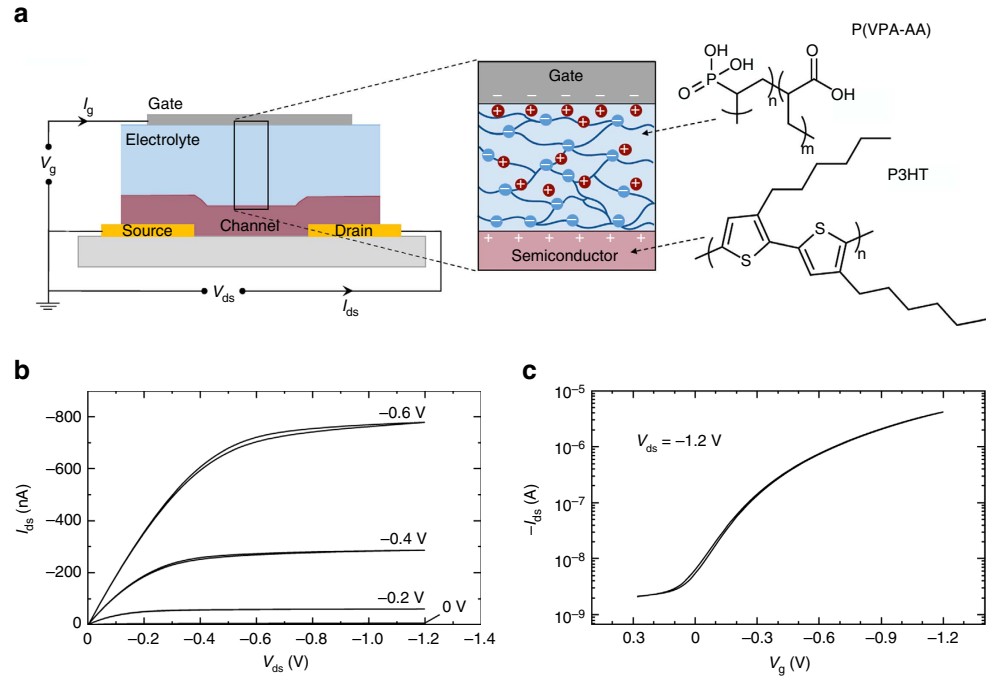

**Figure 1 | Electrical characteristics of an electrolyte-gated transistor.** Schematic diagram of an electrolyte-gated transistor and illustration of the channel charge and ion distribution within the electrolyte layer. The chemical structures of P3HT and P(VPA-AA) are also reported (**a**). Representative output characteristics of the P3HT-based electrolyte-gated transistor (Ti gate metal, $L = 10\,\mu m$, $W = 1\,mm$) (**b**). Corresponding transfer characteristics at $V_{ds} = -1.2\,V$ (**c**).

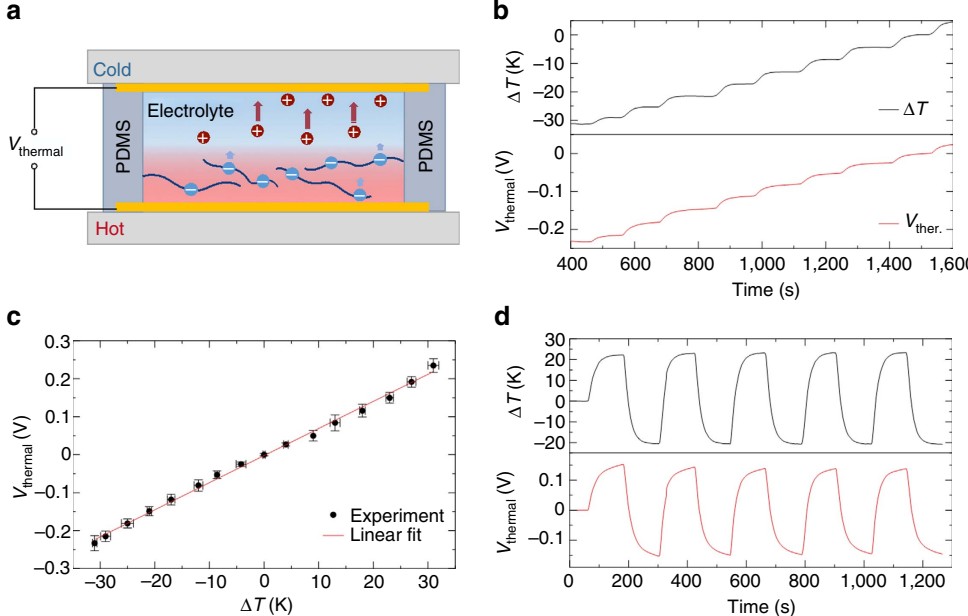

**Figure 2 | Electrical characteristics of the ionic thermoelectric voltage generators.** Schematic diagram of the ITESC. Positive charges are mobile $Na^+$ cations, while negative charges are immobile alkoxylate and carboxylate anions (**a**). Measured $V_{thermal}$ and $\Delta T$ between the two Au electrodes (**b**). Linear fitting of $V_{thermal}$ with different $\Delta T$ values. For each point, $\Delta T$ is fixed until $V_{thermal}$ was stable. The slope of the linear fit to the data gives a Seebeck coefficient of $7\,mV\,K^{-1}$. (**c**) The error bar represents the fluctuation of $\Delta T$ and $V_{thermal}$ at each point. Switching characteristics of the device showing a response time of 25 s (50 s) for 70% (90%) of the saturated voltage (**d**).

composed of polymeric anions that are mostly immobile and mobile $Na^+$ cations, as shown in Fig. 2a. The PEO-NaOH solution is then injected into a small cylindrical cavity (1.5 mm thick, diameter of 10 mm), comprising two planar gold electrodes (see 'Methods' section for further details). When a temperature different $\Delta T$ is applied between the two electrodes, the more mobile $Na^+$ cations diffuse fast towards the cold side, while uncompensated less mobile alkoxylate and carboxylate anions remain at the hot side. This generates a high Soret-induced open voltage between the two electrodes. At each given $\Delta T$, the

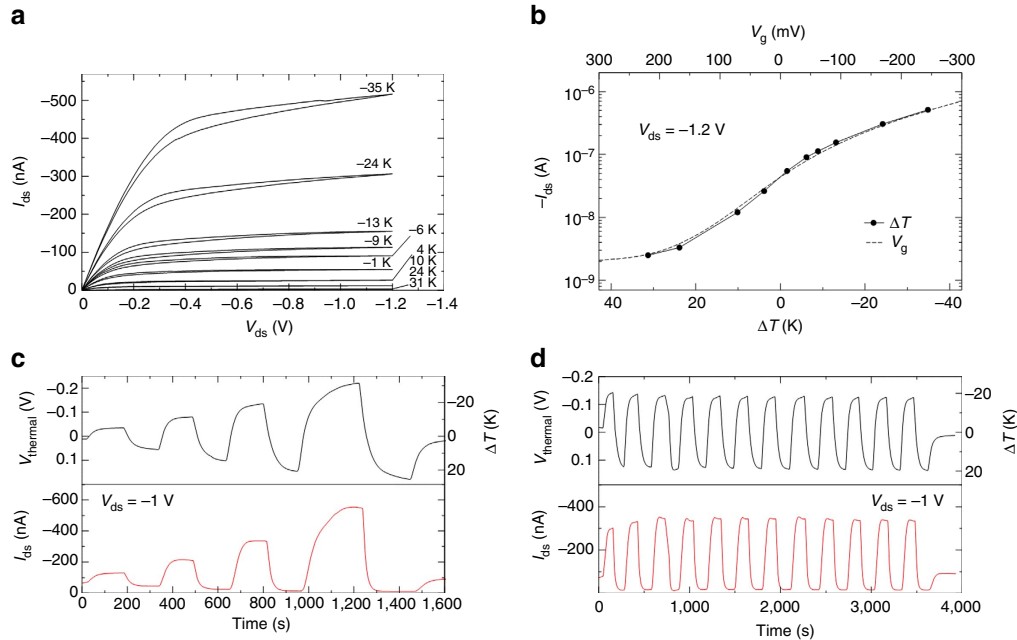

**Figure 3 | Electrical characteristics of the heat-gated transistors.** Output characteristics at different fixed gating $\Delta T$ (Ti gate, channel length $L = 2\,\mu m$) (**a**). Transfer characteristics with channel length $= 2\,\mu m$ (**b**). Output current tracking with the variation of $\Delta T$ (**c**). Multi circles operation of the heat-gated transistor (**d**).

corresponding thermal voltage can be acquired. In Fig. 2b, the thermovoltage at different $\Delta T$ was studied in the temperature range from 15 to 45 °C. For a fixed $\Delta T$, the voltage was measured after the thermovoltage is stabilized for 5 min. The results are presented in Fig. 2c and demonstrate that $V_{thermal}$ varies linearly with $\Delta T$, exhibiting a remarkable Seebeck coefficient of $7\,mV\,K^{-1}$. The switching time of the ITECS is of 25 s and 50 s at 70 and 90% of the response, respectively (Fig. 2d). This Seebeck coefficient is well within the high-performance range reported previously for PEO-NaOH-based ITESC, assuring that devices in this study are representative of the high-performance material. Hence, ionic polymer solutions can provide remarkably high and stable $V_{thermal}$ over a long period of time, facilitating the function of gating a transistor.

**Heat-gated transistors.** Having characterized the thermoelectric properties of PEO-NaOH and the electrical properties of P3HT-based electrolyte-gated transistors, we combine the two in an ionic thermoelectric-gated transistor. This heat-gated transistor is operated by integrating the two devices in series (details can be found in Supplementary Fig. 2): the working electrode of the thermoelectric generator is connected to the transistor gate electrode, while the second ITESC electrode is connected to the source electrode (grounded). Figure 3a shows the typical output characteristics of a P3HT-based ionic thermoelectric-gated transistors recorded at different $\Delta T$. The drain current is measured after $V_{thermal}$ stabilizes for each $\Delta T$. As shown in Fig. 3a, a large heat modulation of the drain current, both in the linear and saturation regimes, is observed when $\Delta T$ is increased from 31 to $-35\,K$. The output curves show proper saturation at high drain biases with negligible hysteresis. A relatively large on-to-off current ratio of more than two orders of magnitude is extracted from the transfer characteristics reported in Fig. 3b (transfer characteristic curves of different channel length are shown in Supplementary Fig. S3). Compared with the transfer characteristics of the same electrolyte-gated transistor recorded when an external gate bias is applied to the gate (dashed line in Fig. 3b),

the variable $\Delta T$ produces a similar transfer curve for the ionic thermoelectric-gated transistor. This demonstrates that ionic thermoelectric effect can efficiently be used to gate the transistor. Note that, although the drain current varies in response to $\Delta T$, it can also be used to measure the absolute value of temperature if the temperature at one side of the ITESC is known (details is shown in Supplementary Fig. 4).

The modulation of the drain current with temperature is also investigated by tracking different $\Delta T$. The temperature of the two electrodes of the ionic thermoelectric generator is switched every 300 s (when $V_{thermal}$ is stable), and $\Delta T$ is increased after every switch. Such a relatively long interval is required to establish a stable $\Delta T$ throughout the device, (see details in Supplementary Fig. 5). Figure 3c clearly shows that the drain current follows the variation of $V_{thermal}$ very closely. Note that the response time of the ionic thermoelectric-gated transistor is not limited by the switching time of the electrolyte-gated transistor, but rather by the response of the thermoelectric component. Indeed, in electrolyte-gated transistors, the EDL at the semiconductor-electrolyte interface forms within only 1–10 ms[26,27], while the response time of the ITESC is 25–50 s (see Fig. 2d and Supplementary Fig. 5). The on current is overall remarkably stable over time as observed by operating different cycles at the same $\Delta T$ (see Fig. 3d). As the time response of the ITESC decreases with the square root of the length of the thermoelectric leg, faster response can be obtained by decreasing the thickness. However, instabilities in $V_{thermal}$ needs to be taken into account when reducing the device dimensions as $\Delta T$ also decreases. Although our heat-gated transistors do not operate at high frequency, they may find applications in important low-frequency range technology, such as e-skin for health monitoring. Indeed, low-frequency changes in skin temperature typically occur in the range of 0.005–0.05 Hz (that is, 20–200 s). These small skin temperature variations (about 1 K) correlate with tissue blood flow and can function as naturally occurring markers for monitoring periodic contraction and dilation of the vessels, that have diagnostic value for conditions such as congestive heart disease and tissue hypoxia[5].

Traditionally, the figure-of-merit that determines the conversion of a modulation in the gate voltage $\Delta V_g$ to a modulation in the drain current $\Delta I_{ds}$, is the transistor's transconductance. The latter is defined as $g_m = \Delta I_{ds}/\Delta V_g$ and it is the main transistor parameter that governs signal amplification. In a similar fashion, we can define a thermal transconductance ($g_{thermal}$) for the ionic thermoelectric-gated transistor as $g_{thermal} = \Delta I_{ds}/S\Delta T$. The transistor's transconductance should be maximized at the minimum required gate voltage in order to detect minute electrical potentials and increase sensitivity[20]. Tuning the electrolyte-gated transistor's transfer characteristic with its maximum transconductance at zero applied $V_g$ has been accomplished, for example, by simply varying the channel geometry and/or through

the use of a different gate electrode material[28]. As in the case of electrolyte-gated transistors[29,30], we found that different gate electrode materials shift the transfer curve in a way that reflects the voltage drop at the gate/electrolyte interface (Fig. 4a). Cu, being a high work function metal electrode ($\Phi_{Cu} = 4.3$ eV), is the most effective at turning the transistor ON at already zero $\Delta T$ (or zero applied $V_{thermal}$), while a larger temperature gradient (high $V_{thermal}$) needs to be applied to Ti, a low work function electrode ($\Phi_{Ti} = 3.8$ eV), to achieve the same effect. The transfer characteristic curves for different channel lengths and stability of Cu gate devices are shown is Supplementary Fig. 6. Accordingly, the transconductance extracted for our ionic thermoelectric-gated transistor with Cu as the gate metal is three times larger than that obtained for equivalent Ti-gated transistors (Fig. 4b). This allows for an appreciable detection of current ($\Delta I_{ds} = 20$ nA) for $\Delta T$ as low as 1 K. Note indeed that to get the same increment in $I_{ds}$ with Ti-gated transistors, a $\Delta T$ of 3.8 K would be required. On the other hand, Ti-gated transistors possess a much higher on/off ratio in the full temperature range considered here (see Supplementary Fig. 7). The gate electrode materials therefore represent an effective route to tune the thermal voltage at which the high transconductance (or high on/off ratio) is achieved. In addition to changing the gate electrode material, $V_{thermal}$ can be enhanced through the same strategy used in traditional $p$-type only thermoelectric generators[31], that is, by adding in series thermocouples made of PEO-NaOH leg and a metal (or conducting polymer) leg of vanishingly small Seebeck coefficient.

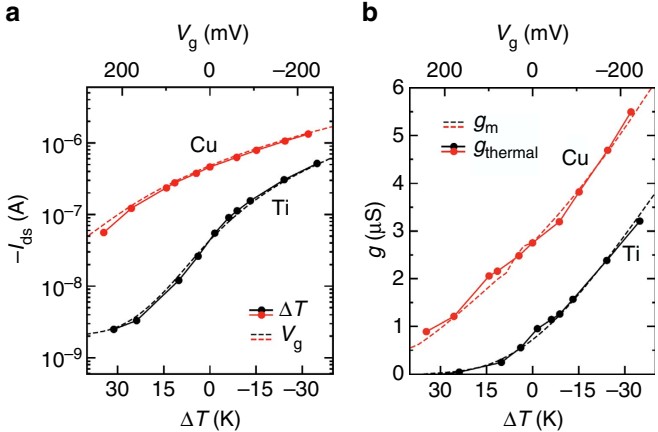

**Figure 4 | Effect of gate metal electrode on heat-gated transistors.** Comparison of the transfer characteristics of ionic thermoelectric-gated transistors comprising Ti and Cu as the gate electrode (**a**) and relative transconductance (**b**).

**Heat-gated inverters**. A simple logic circuit one could build with an ionic thermoelectric-gated transistor is a resistor-load inverter. Such a circuit is shown in the diagram in the inset of Fig. 5, in which the channel is connected to a battery through a series resistor, across which the output voltage is measured. In this configuration, a change in the temperature gradient applied to the

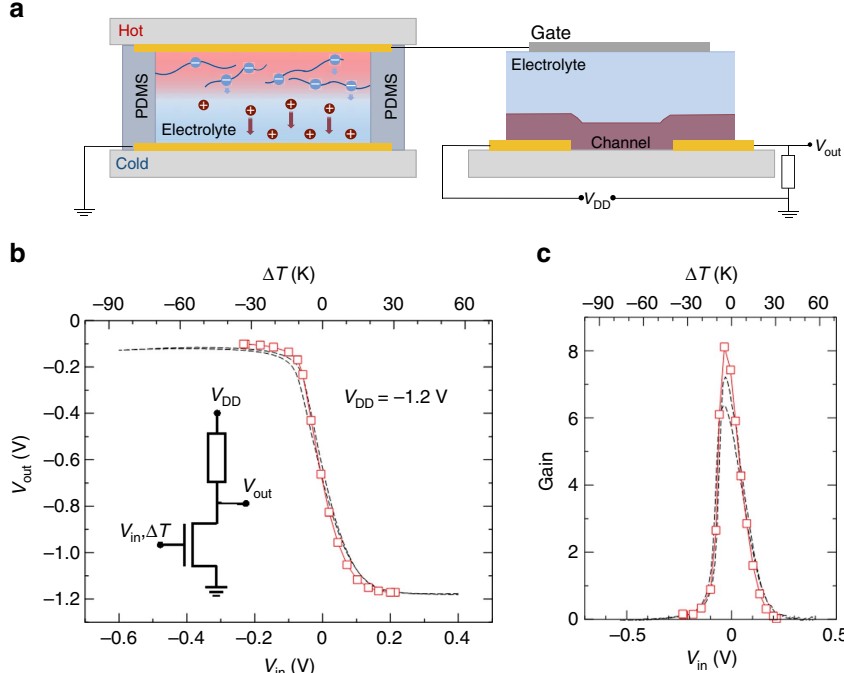

**Figure 5 | Electrical characteristics of heat-gated inverters.** Schematic diagram of the ionic thermoelectric-gated inverters (**a**). Output voltage response ($V_{out}$-$\Delta T$ and $V_{out}$-$V_{in}$) of a resistor-loaded inverter based on a heat-gated transistor for $V_{DD} = -1.2$ V and temperature gradient $-32$ K $< \Delta T < 30$ K or supply input voltage $-0.6$ V $< V_{in} < 0.4$ V. The inset shows the related circuit diagram (**b**). Corresponding signal gain of the inverter for different temperature gradient $\Delta T$ and supply input voltage $V_{in}$ (**c**).

gate is manifested as a voltage drop across the drain resistor. We fabricated a unipolar inverter by connecting our ionic thermo-electric-gated transistor to a $10\,M\Omega$ external load. The value of the resistor was selected so that enough current flows through the load to drop most of the supply voltage ($V_{DD}$) across it (considering $I_{ds} = \sim 10^{-5}\,V$ at $V_g = -1.2\,V$ for channel length $L = 2\,\mu m$, channel width $W = 1\,mm$). The temperature gradient-output voltage characteristic ($\Delta T$-$V_{out}$) is plotted in Fig. 5b. The output voltage switches from a low state ($-1.2\,V$) to high state ($0\,V$) when the temperature gradient is swept from $\Delta T = 30$ to $-32\,K$. The voltage transfer characteristic shows a 1.1 V output swing, which makes use of 92% of the available supply voltage ($V_{DD} = -1.2\,V$). The switching takes place near $\Delta T = 0\,K$ and the inverter is completely switched within only 20 K of $\Delta T$ variation, with a maximum gain ($dV_{out}/dS\Delta T$) as high as 8 (Fig. 5c). Remarkably, the inverter characteristics closely resemble those obtained by gating the inverter with an external gate bias. The inverter is the most basic element in circuits used to construct amplifiers and logic gates. Our heat-gated inverter's characteristics comply with circuit design requirements such as wide output swing and proper gain, which may be tailored by choice of the load and driver gate electrode materials, depending on the design specification, in order to realize larger and more complex integrated circuits.

## Discussion

The high ionic Seebeck voltage of the polymer electrolyte is about 100 times larger than the typical Seebeck voltage of electronic thermoelectric materials. This enable the use of one single ther-moelectric leg-based device to reach enough voltage to switch the transistor. To reach the same voltage with an electronic thermoelectric material, a thermoelectric module composed of 100 legs would be required. This emphasizes the unique feature of ionic thermoelectric materials: it is possible to couple one thermoelectric leg per transistor and design an array of smart sensing pixel. The resolution and sensitivity of such device can be dramatically enhanced by connecting more legs electrically in series but thermally in parallel.

The miniaturization of the ITESC could allow to reach the same thermoelectric leg density as that of state of the art thermopiles (hundreds to thousands per $cm^2$) to boost the voltage responsivity from 100 to $10^4\,V/W$ thus reaching the high sensitivity of pyroelectric detectors. Another way to see the impact of replacing the electronic thermoelectric material with the ionic thermoelectric polymer in a thermopile is that the device would lead to the same voltage by decreasing the number of thermocouples and the size of the device by two order of magnitudes. This would enable thermography by arranging tiny thermopiles in a two-dimensional array. Also, the direct coupling of an organic transistor to an ionic thermopile would enable at least a gain in sensitivity by one order of magnitude, as suggested by the gain obtained in our thermoelectric inverter. It should also be noted that the possibility to control charge accumulation in the transistor channel by inducing different $\Delta T$ is not feasible with conventional inorganic or polymeric dielectric gate insulators since the gate and drain-source driving voltages are relatively much higher in traditional transistors.

In summary, we develop an ionic thermoelectric-gated organic transistor. Such a transistor converts a modulation in $\Delta T$ to a modulation in the drain current $\Delta I$. Since the operating voltage of our electrolyte-gated transistors is of the same order of magnitude as the variation of the thermal voltage generated by the ITESC, we demonstrate that it is possible to tune the transistor output current over more than two orders of magnitude. Note that in principle, the ionic thermoelectric material could also directly replace the electrolyte of the transistor. We demonstrate for the first time that heat signal can act as input for logic circuits, opening up new possibility to couple electrical conductivity with thermoelectricity, in the new field of thermoelectronics.

Although we have only demonstrated the concept of heat sensing by direct physical contact between heat sink and sensor, our concept would also hold for heating through infrared radiation, thus positioning our finding in the field of infrared photonics. In that context, the gating of a low-voltage organic transistor with an ionic thermoelectric device/ionic thermopile (single or multiple thermocouples) combines several unique features for advances in thermography: addressability of a heat-sensing pixel in a two-dimensional array, high sensitivity that could compete with pyroelectric detection, constant signal under illumination. Hence, the ionic thermoelectric-gated transistor is likely to have important applications in thermometry, thermography, smart sensors, human-organic electronic technology interfacing (that is, e-skin) and active flexible electronics.

## Methods

**Materials.** Regioregular poly(3-hexylthiophene) (P3HT, Sigma-Aldrich) was dissolved in 1,2-dichlorobenzene ($10\,mg\,ml^{-1}$) and filtered with a $0.2\,\mu m$ polytetrafluoroethylene syringe filter. Poly(vinylphosphonic acid-co-acrylic acid) (P(VPA-AA)) was purchased from Rhodia, and dissolved in a mixture of 1-propanol and deionized water ($40\,mg\,ml^{-1}$, solvent ratio of 4:1). The polyelectrolyte solution was then filtered with a $0.2\,\mu m$ nylon syringe filter.

**Transistor fabrication.** Interdigitated source and drain electrodes (3-nm-thick Ti and 25-nm-thick Au, patterned by photolithography) were prepared on pre-cleaned corning glass substrates. The substrates were cleaned before use by means of deionized water, acetone and isopropanol. The semiconductor layer was spin-coated from warm solution at 2,000 r.p.m. for 30 s, giving a 30-nm-thick film. The films were then annealed at $120\,°C$ for 30 min under nitrogen. The P(VPA-AA) solution was spin-coated at 2,000 r.p.m. for 60 s and dried on a hot plate under vacuum at $120\,°C$ for 120 s, resulting in a film thickness of about 130 nm. Top gate electrode with a thickness of 80 nm were formed by thermal evaporating various metals through a Ni shadow mask (Tecan Ltd.).

**TEG fabrication.** Two gold thermistors ($30\,\mu m$ width) were patterned on glass substrate by photolithography. A $1$-$\mu m$-thick $Si_3N_4$ layer is deposited by chemical vapour deposition as an insulating layer. Two round gold electrodes were thermally evaporated (10 mm diameter) on the glass substrate to define the active device area. The prepared glass substrates and a PDMS spacing layer (1.5 mm thickness) were treated for 3 min with UV plasma, contacted and baked at $70\,°C$ in oven for 30 min (cavity volume $= 0.0785\,cm^3$). PEO-NaOH solution (3 wt%) was prepared following the procedure early reported in ref. 16, and injected into the chamber cavity.

**Device characterization.** The transistors were characterized using a semi-conductor parameter analyser (Keithley 4200-SCS). Impedance measurements were carried out with an Alpha high-resolution dielectric analyser (Novocontrol GmbH). A $V_{AC}$ of 0.001 V at a frequency of 1 kHz was applied, while a $V_{DC}$ was swept. The equivalent circuit model used to extract the effective capacitance can be found in ref. 30. A nanovoltmeter (Keithley Instruments, Inc., model 1282 A) was used to measure the thermal voltage of the thermoelectric generator, while a Keithley 2400 was used to measure the resistance of the thermistor simultaneously.

**Data availability.** The authors declare that the data supporting the findings of this study are available within the paper and its supplementary information files.

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

## Acknowledgements

The authors acknowledge the European Research Council (ERC-starting-grant 307596), the Knut and Alice Wallenberg foundation (project 'Tail of the sun'), the Swedish Foundation for Strategic Research (Synergy project), the Swedish Energy Agency and the Advanced Functional Materials Center at Linköping University. S.F. also gratefully acknowledges funding by the Swedish Governmental Agency for Innovation Systems (No. 2015-04859).

## Author contributions

X.C. and S.F. conceived and designed the project. D.Z. and S.F. fabricated the devices and analysed the data. D.Z. fabricated and tested the ionic thermoelectric generators. S.F. fabricated and tested the electrolyte-gated transistors. M.B. supervised the project. D.Z. S.F. and X.C. wrote the paper.

## Additional information

**Competing financial interests:** The authors declare no competing financial interests.

