## [Peer Review File · Nature Communications]

Reviewers' comments:

Reviewer #1 (Remarks to the Author):

- Who will be interested in reading the paper, and why?

The paper paves the way to a new device that can be realized now due to the fact that extremely high Seebeck coefficients can be obtained using 11 mV/K being sufficient creating a channel in a OFET. The others explain that due to the need of such a device other methods such as pro-electric devices have been tested in future, however the possibility of gating a OFET by the Seebeck effect can be considered as a true breakthrough. This effect is to the knowledge of the reviewer, indeed, published for the first time in this paper

- What are the main claims of the paper and how significant are they?
 1. The claims is pioneering work presenting the first Electrolyte-gated transistor whereby due to extremely high Seebeck coefficients sufficient charge can be accumulated creating a channel.
 2. Based on the concept that was initiated by the authors in a previous paper (Ref. 36) thermoelectric supercapacitors, allowing to be loaded just by exposing the device to a heat flow.
- Is the paper likely to be one of the five most significant papers published in the discipline this year?

Yes, I can easily imagine that.

- How does the paper stand out from others in its field?

The paper is truly outstanding; its value is based on the direct implementation of an historically record of Seebeck effect directly into a device.

- Are the claims novel? If not, which published papers compromise novelty?

Yes

- Are the claims convincing? If not, what further evidence is needed?

Yes

- Are there other experiments or work that would strengthen the paper further?

The authors are high experts in the field and did the straightforward experiments and conclusions.

- How much would further work improve it, and how difficult would this be? Would it take a long time?
-

Not needed

- Are the claims appropriately discussed in the context of previous literature?

Yes

- If the manuscript is unacceptable, is the study sufficiently promising to encourage the authors to resubmit?
-

The manuscript is sufficiently acceptable for describing the claims and convincing the scientific community.

- If the manuscript is unacceptable but promising, what specific work is needed to make it acceptable?
-

Not needed.

- Is the manuscript clearly written?

It is well written for true experts, it is difficult to understand by non-experts

- If not, how could it be made more clear or accessible to nonspecialists?

There is an urgent need for better sketches of the device and how it is characterized; I feel that the in-figure 1 added schematic sketch of the transistor is a poor illustration for all what is concluded in the paper for it. A further indication on electronic circuitry is (in a minimum manner) sketched in Fig. 5 is too poor for this paper. I would like to see a combine high quality sketch showing what is V_{DC} , I_{DC} , g_m and others.

Furthermore the process steps for the devices Transistor and TEGs are very poorly described between the lines 298 and 314. This must, for better understanding of non experts, supported by a sketch or a process-flow sequence.

- Would readers outside the discipline benefit from a schematic of the main result to accompany publication?

Not, as already summarized above.

- Could the manuscript be shortened? (Because of pressure on space in our printed pages we aim to publish manuscripts as short as is consistent with a persuasive message.)

The information density is already high, what takes a lot of space is the general explanation of the problem and the motivation to solve it, more shortening might be justified for expert readers but for non-experts it would be rather impossible to follow.

- Should the authors be asked to provide supplementary methods or data to accompany the paper online? (Such data might include source code for modelling studies, detailed experimental protocols or mathematical derivations.)

The literature list in combination for the explanation in the paper is sufficient.

- Have the authors done themselves justice without overselling their claims?

No

- Have they been fair in their treatment of previous literature?

Yes

- Have they provided sufficient methodological detail that the experiments could be reproduced?

No, this could be improved.

- Is the statistical analysis of the data sound, and does it conform to the journal's guidelines?

Yes.

- Are the reagents generally available?

Yes

- Are there any special ethical concerns arising from the use of human or other animal subjects?

No.

Reviewer #2 (Remarks to the Author):

Ionic Thermoelectric Gating Organic Transistors by D. Zhao et al.

The authors have developed an electrolyte-gated polymeric FET transistor coupled to a charged ionic thermoelectric supercapacitor where a voltage difference is created by the electrode double layer due to a temperature difference. This setup allows a temperature modulation to be converted into a modulation of the drain current. The aim of this thermoelectric device is to measure temperature with a high sensitivity.

The work is interesting and well written. The authors characterize separately the electrical properties of a P3HT-based electrolyte-gated transistor and the ionic polymeric thermoelectric supercapacitor module that shows a large Seebeck coefficient of about 7 mV/K. This thermoelectric voltage is sufficient to switch the transistor. In a second step they combine the two parts to form a thermoelectric-gated transistor tested at different ΔT values or under transient ΔT . They show that the drain current is modulated by the temperature gradient and that it follows the variation of the temperature relaxation.

By using a resistor-loaded inverter, they were able to control the switching from a low state to a high state when the temperature gradient is changed from $\Delta T = 30\text{K}$ to -30K with a switching almost complete within a 20 K interval.

This setup is promising for temperature measurement and for miniaturization. However, it could be interesting to know the response time of the setup that would depend mostly on the formation of the electrical double layer and the diffusion of the ions across the thickness of the supercapacitor. This might cause a problem for high frequency temperature measurements. Moreover, absolute temperature measurement would be difficult to perform because the device responds only to a ΔT difference between the electrodes of the supercapacitor.

The manuscript certainly deserves publication in Nature Communications particularly for its originality and interest, after some minor corrections.

Minor Comments:

Line 83, p. 5: Ampullae of Lorenzini. I think that the interpretation using the thermoelectric effect of the gel electrolyte has been shown to be wrong by other authors. The authors should correct/modify this part.

Line 101, p. 6. Please define TEG

Line 145, p. 8. "... Experimental Section..." should be "Methods"

Line 261, p. 14. "too" \diamond "to"

Reviewer #3 (Remarks to the Author):

The paper „Ionic thermoelectric gating organic transistors“ from D. Zhao et al. report on an ionic thermoelectric-gated organic electrochemical transistor. In fact the ionic thermoelectric is based on an electrolyte (PEO-NaOH) working as the dielectric in a supercapacitor showing a very high Seebeck coefficient (7 mV/K), which was already reported in a former publication of the group. In the recent manuscript they couple the ITESC capacitor to the gate of a standard electrolyte-gate transistor and thus could show a complete ΔT -induced switching of the drain current and even of a resistive-load inverter

Without any doubt, this investigation opens a novel route towards the realization of heat-controlled organic transistors and even organic amplifiers and it is not only important for the field of organic electronics, but (due to the large S-coefficient) could also be used as an alternative to the SoA electronic thermoelectric materials: just one leg with a smaller footprint would enable to fabricate

large-area arrays for T-sensitive e-skin etc...

Therefore and due to the very good results the paper is recommended for publication after some modifications w.r.t the following suggestions are done:

1. The paper is clearly written, however, here and there it would be beneficial to proofread the manuscript by a native speaker
2. Any claim of novelty like "To the best of our knowledge, this is the first demonstration of thermoelectric gating of a transistor" is questionable. Leave it to the reader to decide on this.
3. The results and the data analysis are very convincing and conclusive, most of the questions are well addressed except the topics listed below:
 - i) The very large reaction time (fall and rise time ~ 100 s !) upon T-cycling has to be discussed in more detail, since it will strongly influence any circuit design of such a device. Is it due to the very limited speed of electrolyte-gated transistors, the time to reach T-equilibrium within the layers or/ and the expected delay in T-response of the ITESC due to slow Na ion motion ? Standard passive thermal IR-detectors or pyroelectric sensors have a reaction time in the range of few s.
 - ii) Moreover, the claimed ΔT -resolution of only 1K for the Cu-gated transistors is not obvious from Fig. 4b. Is it extrapolated from the slope of the curve or is it directly measured via small ΔT -steps? How would the I_d -modulation look like for such a small T-variation ?
 - iii) Finally, the overall T-range needed to fully switch the transistor from On-to-Off or e.g. to switch the inverter is quite large ~ -30 K to 30K. Is that of any practical use ?

REVIEWERS' COMMENTS:

Reviewer #2 (Remarks to the Author):

The authors have clarified the manuscript according to my comments. The manuscript should be published in Nature Communications after correction of Fig. S4.a.

I think that Fig S4.a is wrong as the Delta T line (red line) does not represent the temperature difference ($T_1 - T_2$). It seems rather to follow the T_1 variation only, without subtracting the T_2 value. By comparison, the plot of Delta T in Fig. S4.b is good, and does reflect ($T_1 - T_2$), T_2 being close to 0.

Answers to the reviewer comments

Reviewer #1:

The paper paves the way to a new device that can be realized now due to the fact that extremely high Seebeck coefficients can be obtained using 11 mV/K being sufficient creating a channel in a OFET. The others explain that due to the need of such a device other methods such as piezoelectric devices have been tested in future, however the possibility of gating a OFET by the Seebeck effect can be considered as a true breakthrough. This effect is to the knowledge of the reviewer, indeed, published for the first time in this paper.

We thank the reviewer for the very positive feedback that gives us the possibility to improve the quality of our manuscript. In the following is a detailed answer to each point raised by the reviewer:

There is an urgent need for better sketches of the device and how it is characterized; I feel that the in-figure 1 added schematic sketch of the transistor is a poor illustration for all what is concluded in the paper for it. A further indication on electronic circuitry is (in a minimum manner) sketched in Fig. 5 is too poor for this paper. I would like to see a combine high quality sketch showing what is VDC, IDC, gm and others. Furthermore the process steps for the devices Transistor and TEGs are very poorly described between the lines 298 and 314. This must, for better understanding of non experts, supported by a sketch or a process-flow sequence.

This is an excellent suggestion. Based on the reviewer comment, we included in the revised version of our manuscript a schematic sketch of the devices, for better understanding of the non-experts.

Here are the modifications introduced:

On page 6:

“Top gate, bottom contact transistors based on regioregular poly(3-hexylthiophene-2,5-diyl) (P3HT) are fabricated using poly(vinylphosphonic acid-co-acrylic acid) (P(VPA-AA)) as the polyanionic electrolyte insulator (Figure 1a). The P(VPA-AA) phosphonic acid groups are strongly acidic (the first acidic constant, pK_{a1} , is about 2.5 ± 0.5 in water), and therefore provide plenty of potentially mobile cations (ca. 8 mmol g^{-1} , considering one dissociated proton per phosphonic acid group). The anionic phosphonate groups are instead virtually immobile, preventing penetration of anions into the conjugated polymer when a negative gate is applied, and thus precluding electrochemical doping of the bulk semiconducting polymer layer”

On page 7 and 8:

Figure 1. Electrical characteristics of an electrolyte-gated transistor. Schematic diagram of an electrolyte-gated transistor and illustration of the channel charge and ion distribution within the electrolyte layer. The chemical structures of P3HT and P(VPA-AA) are also reported (a). Representative output characteristics of the P3HT-based electrolyte-gated transistor (Ti gate metal, $L = 10 \mu\text{m}$, $W = 1 \text{mm}$)

On page 8:

Figure 2. Electrical characteristics of the ionic thermoelectric voltage generators. Schematic diagram of the ionic thermoelectric supercapacitors (ITESC). Positive charges are mobile Na^+ cations, while negative charges are immobile alkoxy and carboxylate anions (a). Measured V_{thermal} and ΔT between the two Au electrodes.

On page 14:

Figure 5. Electrical characteristics of heat-gated inverters. Schematic diagram of the ionic thermoelectric-gated inverters (a).

Reviewer #2:

COMMENTS:

The authors have developed an electrolyte-gated polymeric FET transistor coupled to a charged ionic thermoelectric supercapacitor where a voltage difference is created by the electrode double layer due to a temperature difference. This setup allows a temperature modulation to be converted

into a modulation of the drain current. The aim of this thermoelectric device is to measure temperature with a high sensitivity.

The work is interesting and well written. The authors characterize separately the electrical properties of a P3HT-based electrolyte-gated transistor and the ionic polymeric thermoelectric supercapacitor module that shows a large Seebeck coefficient of about 7 mV/K. This thermoelectric voltage is sufficient to switch the transistor. In a second step they combine the two parts to form a thermoelectric-gated transistor tested at different ΔT values or under transient ΔT . They show that the drain current is modulated by the temperature gradient and that it follows the variation of the temperature relaxation.

By using a resistor-loaded inverter, they were able to control the switching from a low state to a high state when the temperature gradient is changed from $\Delta T = 30\text{K}$ to -30K with a switching almost complete within a 20 K interval.

The manuscript certainly deserves publication in *Nature Communications* particularly for its originality and interest, after some minor corrections.

We thank the reviewer for the very positive comments and for considering our manuscript appropriate to *Nature Communications*. In the following is a detailed answer to each point raised by the reviewer:

This setup is promising for temperature measurement and for miniaturization. However, it could be interesting to know the response time of the setup that would depend mostly on the formation of the electrical double layer and the diffusion of the ions across the thickness of the supercapacitor. This might cause a problem for high frequency temperature measurements.

The response time of our combined thermoelectric/transistor device is limited by the thermoelectric component. Indeed, from previous studies on electrolyte-gated transistors, the electric double layer at the semiconductor-electrolyte interface forms within only 1-10 milliseconds [*Adv. Mater.* **2007**, *19*, 97-101, *PNAS* **2011**, *108*, 15069-15073]. We plotted the response time of the thermoelectric leg in Fig. 2c (added below, and integrated in the new version of the manuscript). It can be seen clearly that the thermodiffusion of ions generates a thermovoltage that is changing slower than the temperature difference. Hence, the thermodiffusion of ions is limiting the response time of the device. The device response times, including 70% and 90% of the thermovoltage variation, is measured to be about 25s and 50s, respectively.

Fig. S5 (a) Measured ΔT across the thermoelectric leg versus time (black dot-dashed line) and the time evolution of V_{thermo} (red dashed line). (b) Detailed zoom in of a).

To better illustrate the thermoelectric response of our device, we report the first derivative of both ΔT and V_{thermo} in the figure below:

Figure S5. c) the derivation of b), shows the changing rate of ΔT and V_{thermo} .

The variation of V_{thermo} follows exactly that of ΔT , with a maximum speed of 5-10 mV/s. These results show that our ionic thermoelectric device is able to efficiently and promptly transfer a variation in ΔT to a variation in V_{thermo} . As the time response of our device decreases with the square root of the length of the thermoelectric leg, faster response can be obtained by decreasing the thickness. However, instabilities in V_{thermal} needs to be taken into account when reducing the device dimensions as ΔT also decreases. We would certainly welcome the ability to operate our devices at high frequency for temperature measurements, however we believe this is beyond the scope of this work. Indeed, although these devices may not be suitable for high-frequency operation yet, they may find applications in important low-frequency range technology, such as e-skin for health monitoring. Indeed, as small low-frequency changes in skin temperature (typically in the range 0.005-0.05 Hz) correlate with tissue blood flow, they can serve as naturally occurring markers for monitoring periodic contraction and dilation of the vessels.

We now discuss this in more details in the main text:

Page 8:

“The switching time of the ITECS is of 25s and 50s at 70% and 90% of the response, respectively (Figure 2d).”

Page 9:

Figure 2. Electrical characteristics of the ionic thermoelectric voltage generators. Switching characteristics of the device showing a response time of 25 s (50s) for 70% (90%) of the saturated voltage (d).

Page 11:

“Such a relatively long interval is required to establish a stable ΔT throughout the device, (see details in Figure S5).”

“Note that the response time of the ionic thermoelectric-gated transistor is not limited by the switching time of the electrolyte-gated transistor, but rather by the response of the thermoelectric component. Indeed, in electrolyte-gated transistors, the EDL at the semiconductor-electrolyte interface forms within only 1-10 milliseconds^{39,41}, while the response time of the ITESC is 25-50s (see Figure 2d and Figure S5)”

“As the time response of the ITESC decreases with the square root of the length of the thermoelectric leg, faster response can be obtained by decreasing the thickness. However, instabilities in V_{thermal} needs to be taken into account when reducing the device dimensions as ΔT also decreases. Although our heat-gated transistors do not operate at high-frequency, they may find applications in important low-frequency range technology, such as e-skin for health monitoring. Indeed, low-frequency changes in skin temperature typically occur in the range of 0.005-0.05 Hz (i.e. 20-200s). These small skin temperature variations (about 1K) correlate with tissue blood flow and can function as naturally occurring markers for monitoring periodic contraction and dilation of the vessels, that have diagnostic value for conditions such as congestive heart disease and tissue hypoxia.”

Moreover, absolute temperature measurement would be difficult to perform because the device responds only to a ΔT difference between the electrodes of the supercapacitor.

It is true that this device will not be able to detect the absolute temperature if both sides are changing. However, in our devices, if the front side of the thermoelectric device is exposed to the temperature change and the back side doesn't have any temperature modulation (only left to air with room temperature). ΔT can be related to the real temperature of the front side, as shown below (also shown as Figure S4.a).

Fig. S4. a) The correlation between measured ΔT changing with real temperature. a) ΔT changing with the real temperature of the front side of the device, while the temperature of the other electrode

is kept constant. The black is the temperature of the front (T_1) side, and black dashed line is the temperature of the other side (T_2), and the red dashed line is ΔT .

Moreover, the strategy to be used is similar to that of commercial thermopile. In those devices, the front side of the thermoelectric device is exposed to the temperature change and the back side is connected to a heat sink in order to keep its temperature constant. Hence, reading the thermovoltage gives directly a measurement of the temperature on the front side.

Similar strategy can be used here and this is displayed in Figure S4.b of the revised manuscript (also shown below): the temperature on one side is changing with time; while the temperature on the other side is kept constant. In this fashion, the absolute temperature value can be directly extracted from the measured ΔT .

Fig. S4. b) ΔT changing with the real temperature of one electrode, while the other electrode is kept constant temperature. During this measurement, the temperature of the front side is heated or cooled, while the other side is leaving open without temperature control, ΔT is measured at the same time.

This part of discussion is added to the main text:

Page 10:

“Note that, although the drain current varies in response to ΔT , it can also be used to measure the absolute value of temperature if the temperature at one side of the ITESC is known (details is shown in Figure S4).”

Minor Comments:

Line 83, p. 5: Ampullae of Lorenzini. I think that the interpretation using the thermoelectric effect of the gel electrolyte has been shown to be wrong by other authors. The authors should correct/modify this part.

We thank the reviewer for giving us the possibility to clarify our statement. We modified in the main text (page5, line 9-12):

“These innervated ampullae form a network of pores and are filled with a gel electrolyte that shows an extraordinary high proton conductivity³⁴. The gel electrolyte was proposed to perform either as a simple ionic conductor³⁵, or as a semiconductor with temperature dependence conductivity and thermoelectric behavior³⁶”

Line 101, p. 6. Please define TEG

Line 145, p. 8. “... Experimental Section...” should be “Methods”

Line 261, p. 14. “too” □ “to”

We modified all mentioned above.

Reviewer #3:

COMMENTS:

The paper „Ionic thermoelectric gating organic transistors” from D. Zhao et al. report on an ionic thermoelectric-gated organic electrochemical transistor. In fact the ionic thermoelectric is based on an electrolyte (PEO-NaOH) working as the dielectric in a supercapacitor showing a very high Seebeck coefficient (7 mV/K), which was already reported in a former publication of the group. In the recent manuscript they couple the ITESC capacitor to the gate of a standard electrolyte-gate transistor and thus could show a complete ΔT -induced switching of the drain current and even of a resistive-load inverter.

Without any doubt, this investigation opens a novel route towards the realization of heat-controlled organic transistors and even organic amplifiers and it is not only important for the field of organic electronics, but (due to the large S-coefficient) could also be used as an alternative to the SoA electronic thermoelectric materials: just one leg with a smaller footprint would enable to fabricate large-area arrays for T-sensitive e-skin etc...

Therefore and due to the very good results the paper is recommended for publication after some modifications w.r.t the following suggestions are done:

We thank the reviewer for the very useful comments giving us the possibility to clarify our statements and to strengthen the final results. In the following we address his/her minor remarks:

1. The paper is clearly written, however, here and there it would be beneficial to proofread the manuscript by a native speaker.

The revised manuscript has been proofread by a native speaker as suggested by the reviewer.

2. Any claim of novelty like “To the best of our knowledge, this is the first demonstration of thermoelectric gating of a transistor” is questionable. Leave it to the reader to decide on this.

- Page 6, we removed the statement “To the best of our knowledge”

- Page 15, we removed the statement “this is the first demonstration of thermoelectric gating of a transistor”

3. The results and the data analysis are very convincing and conclusive, most of the questions are well addressed except the topics listed below:

i) The very large reaction time (fall and rise time ~ 100 s !) upon T -cycling has to be discussed in more detail, since it will strongly influence any circuit design of such a device. Is it due to the very limited speed of electrolyte-gated transistors, the time to reach T -equilibrium within the layers or/and the expected delay in T -response of the ITECS due to slow Na ion motion ? Standard passive thermal IR-detectors or pyroelectric sensors have a reaction time in the range of few s.

As stated above in the response to reviewer #2, the response time of our thermoelectric/transistor device is limited by the thermoelectric component. Previous studies on electrolyte-gated transistors showed in fact that the electric double layer at the semiconductor-electrolyte interface forms within only 1-10 milliseconds [Adv. Mater. 2007, 19, 97–101, PNAS 2011, 108, 15069–15073]. Figure 2c of the revised manuscript shows that thermodiffusion of ions limits the response time of our device, which is measured to be about 50s for a thermovoltage variation of 90%. These devices may find applications in e-skin technology for health monitoring. Low-frequency changes in skin temperature (typically in the range 0.005–0.05 Hz) have been reported to correlate in fact with tissue blood flow, so they can serve as naturally occurring markers for monitoring periodic contraction and dilation of the vessels.

As the time response of our device decreases with the square root of the length of the thermoelectric leg, faster response can be obtained by decreasing the thickness. However, instabilities in V_{thermal} needs to be taken into account when reducing the device thickness as ΔT also decreases.

We now discuss this in more details in the main text:

Page 8:

“The switching time of the ITECS is of 25s and 50s at 70% and 90% of the response, respectively (Figure 2d).”

Page 9:

Figure 2. Electrical characteristics of the ionic thermoelectric voltage generators. Switching characteristics of the device showing a response time of 25 s (50s) for 70% (90%) of the saturated voltage (d).

Page 11:

“Such a relatively long interval is required to establish a stable ΔT throughout the device, (see details in Figure S5).”

“Note that the response time of the ionic thermoelectric-gated transistor is not limited by the switching time of the electrolyte-gated transistor, but rather by the response of the thermoelectric component. Indeed, in electrolyte-gated transistors, the EDL at the semiconductor-electrolyte interface forms within only 1-10 milliseconds^{39,41}, while the response time of the ITECS is 25-50s (see Figure 2d and Figure S5)”

“As the time response of the ITESC decreases with the square root of the length of the thermoelectric leg, faster response can be obtained by decreasing the thickness. However, instabilities in V_{thermal} needs to be taken into account when reducing the device dimensions as ΔT also decreases. Although our heat-gated transistors do not operate at high-frequency, they may find applications in important low-frequency range technology, such as e-skin for health monitoring. Indeed, low-frequency changes in skin temperature typically occur in the range of 0.005-0.05 Hz (i.e. 20-200s). These small skin temperature variations (about 1K) correlate with tissue blood flow and can function as naturally occurring markers for monitoring periodic contraction and dilation of the vessels, that have diagnostic value for conditions such as congestive heart disease and tissue hypoxia.”

ii) Moreover, the claimed ΔT -resolution of only 1K for the Cu-gated transistors is not obvious from Fig. 4b. Is it extrapolated from the slope of the curve or is it directly measured via small ΔT -steps? How would the I_d -modulation look like for such a small T -variation?

This is an excellent comment and we thank the reviewer for giving us the possibility to clarify our statement. Cu-gated transistors show larger transconductance compared to Ti-gated transistors. As the thermal transconductance is defined as $\Delta I_d/S\Delta T$, the larger the transconductance the smaller the ΔT that can be detected. Figure 4 shows that, for ΔT as small as only 1K, the variation in drain current ($\Delta I_{ds} = 20$ nA) for our ionic thermoelectric-gated transistor with Cu as the gate metal is three times larger than that of equivalent Ti-gated transistors ($\Delta I_{ds} = 6$ nA).

The main text is modified on page 12:

“Accordingly, the transconductance extracted for our ionic thermoelectric-gated transistor with Cu as the gate metal is three times larger than that obtained for equivalent Ti-gated transistors (Figures 4b). This allows for an appreciable detection of current ($\Delta I_{ds} = 20$ nA) for ΔT as low as 1K. Note indeed that to get the same increment in I_{ds} with Ti-gated transistors, a ΔT of 3.8K would be required.”

iii) Finally, the overall T -range needed to fully switch the transistor from On-to-Off or e.g. to switch the inverter is quite large \sim -30K to 30K. Is that of any practical use?

We agree with the reviewer that T -range needed to fully switch the transistor from on-to-off is relatively large. However, we aimed to demonstrate here that heat-gated circuits can be fabricated by means of very simple structure (one single ionic thermoelectric leg) as a proof of concept. Based on this result, one can build up more complicated device, e.g. multiple ionic thermoelectric leg connected in series, so that the switch temperature range can be reduced for practical use. This is now discussed in the revised manuscript.

The main text is modified on page 12:

“The resolution and sensitivity of such device can be dramatically enhanced by connecting more legs electrically in series but thermally in parallel.”

Reviewer #2 (Remarks to the Author):

The authors have clarified the manuscript according to my comments. The manuscript should be published in Nature Communications after correction of Fig. S4.a.

I think that Fig S4.a is wrong as the Delta T line (red line) does not represent the temperature difference ($T_1 - T_2$). It seems rather to follow the T_1 variation only, without subtracting the T_2 value. By comparison, the plot of Delta T in Fig. S4.b is good, and does reflect ($T_1 - T_2$), T_2 being close to 0.

Thanks for the reviewer for accept our revision and recommend this paper to by publish in Nature Communication. We agree that is confusing. The scale bar we used in the previous version was tuned to make delta follow T_2 . Actually delta T is the subtraction between T_1 and T_2 . It follows the changing of T_1 because T_2 is dependent on T_1 also, as shown in the new Supplementary Figure 4a:

Supplementary Figure 4. The correlation between measured ΔT changing with real temperature. a) ΔT changing with the real temperature of the front side of the device, while the temperature of the other electrode is kept constant. The black is the temperature of the front (T_1) side, and black dashed line is the temperature of the other side (T_2), and the red dashed line is ΔT